# Effects of prenatal lead, mercury, cadmium, and arsenic exposure on children's neurodevelopment in an artisanal small-scale gold mining area in Northwestern Tanzania using a multi-chemical exposure model

**Elias C. Nyanza**[1]*, **Raphael J. Mhana**[2], **Moses Asori**[3], **Deborah S.K. Thomas**[3], **Agapiti P. Kisoka**[4]

1 Department of Environmental, Occupational, and Research GIS, School of Public Health, Catholic University of Health and Allied Sciences, Mwanza, Tanzania, 2 Reproductive and Child Health Unit, Mwanza Regional Health Services, Mwanza, Tanzania, 3 Department of Geography and Earth Sciences, University of North Carolina at Charlotte, Charlotte, North Carolina, United States of America, 4 Social and Environmental Development, Kahama, Tanzania

* elcnyanza@gmail.com

## Abstract

An estimated 250 million children under five fail to reach their cognitive development potential in Africa. Growing evidence suggests reduced neurodevelopments for children from environmental exposures, yet research on this topic in Sub-Saharan Africa remains limited. This study examined the effects of multi-chemical prenatal exposure to heavy metals on developmental milestones for children aged 3–4 in artisanal and small gold mining areas in northwestern Tanzania. This longitudinal follow-up study of children whose mothers were enrolled in the *Tanzania Mining and Health Cohort* in Geita District in 2015 were assessed at 3–4 years of age for the current study between June 2019 – June 2020. Developmental outcomes (cognitive, social, motor, and language skills) were assessed using the Malawi Developmental Assessment Tool (MDAT). A quantile g-computation model evaluated the linkage between multi-chemical exposures and developmental milestones. Of the 310 children in the follow-up, a majority had at least one form of developmental impairment (50.7%; n = 157) across four domains: gross motor (20.3%), fine motor (23.9%), language (28.3%), and social skill (16.2%). Increased Pb, Hg, Cd, and As exposure jointly reduced gross motor by 17.78% (aPR = 0.822; 95% CI: 0.6994, 0.966). Joint exposure to these heavy metals decreased language ability by 55.36% (aPR = 0.446; 95% CI: 0.313, 0.636) and decreased general developmental milestones by 13.36% (aPR = 0.866; 95% CI: 0.747,1.005). However, the combined effect on the fine motor (aPR = 0.943; 95% CI: 0.754, 1.180) and social skills 6.71% (aPR = 1.067; 95% CI: 0.694, 1.641) were not statistically significant. Exposure to heavy metals while in utero reduced children neurodevelopmental milestones at 3–4 years of age. The

**Data availability statement:** De-identified data and detailed information regarding the participants are available, See S1_Data.

**Funding:** Partial funding for the fieldwork and laboratory work was received from the Department of Environmental and Occupational Health at the Catholic University of Health and Allied Sciences to ECN & the Department of Geography and Earth Sciences, University of North Carolina at Charlotte to MA. The funders had no role in the study design, data collection and analysis, decision to publish, or preparation of the manuscript.

**Competing interests:** The authors have declared that no competing interests exist.

cumulative impact of Pb, Hg, Cd, and As was significant for gross motor, language ability, and general impairment. The independent effects of Pb and Hg were amplified beyond what would be expected under the additive assumption with Cd and As, suggesting synergistic effects.

## 1. Introduction

Exposure to chemicals contributes to an estimated 4.9 million deaths and 86 million disabilities globally every year, representing approximately 25% of the disease burden worldwide [1]. A 2007 study estimated that 200 million children under five in Sub-Saharan Africa and South Asia did not reach their full cognitive development potential, a figure which increased to about 250 million in 2017 [1,2]. From a life course perspective, developmental impairment in early childhood diminishes future educational and occupational opportunities and negatively impacts social and economic mobility [3,4]. By extension, cognitive decline across large swaths of the population restricts the potential for regional and national development [1,4,5].

Environmental exposure in utero and the early years of life increases the risk for physical- and mental health challenges later in adulthood [3,6]. Recent evidence reveals that even the previous generation's exposure can have negative health consequences in later years [7,8]. Exposure to chemicals in utero and early childhood has been associated with neurocognitive and psychomotor developmental impairments in earlier and later childhood [7–10]; with an estimated 25% of early childhood developmental impairments attributed to environmental exposures [7]. In particular, the adverse health effects of arsenic (As), cadmium (Cd), lead (Pb), and mercury (Hg) are well documented even at low doses in early fetal life [7,8,11–13], although the burden is not as well studied in Sub-Saharan Africa.

Exposure to non-essential metallic elements is shaped by human activities, including artisanal and small-scale gold mining (ASGM), which is relatively common in most African countries [14–16]. ASGM activities are a primary source of environmental pollution/contamination, particularly Hg as a result of gold amalgamation, and As, Pb, and Cd as constituents of most gold ore [15,17,18]. Studies conducted in ASGM areas in northwestern Tanzania revealed that soil, water sources, cassava leaves, and roots are contaminated with several chemical elements, including As, Hg, Pb, and Cd [15,19]. Human exposure to these heavy metals occurs through inhalation of polluted air, drinking contaminated water, and consuming food/crops prepared/grown in contaminated soil or irrigated with contaminated water [15,19].

Exposure to toxic chemical elements is cumulative and compounding [20–22]. This may induce synergistic or intensifying health risks beyond an additive effect. For instance, even though Pb reduced the mental development index among children, the effect intensified with Hg and Cd [23]. A similar synergistic effect of Pb in the presence of Hg and polychlorinated biphenyls is reported between neurophysiological disorders and Pb exposure among children in Canada [24]. While chemical-specific studies linking one chemical to a health outcome have provided helpful information to inform regulatory policies, such studies fail to fully reveal the nature of multi-chemical

exposures when individuals are exposed to multiple chemicals simultaneously or sequentially [25]. Improved models that more fully capture the simultaneity of heavy metals exposures can provide more robust evidence for policy.

Pregnant women in artisanal and small-scale gold mining (ASGM) areas are exposed to relatively high levels of chemicals [7,25,26]. Evidence indicates dose-related associations between exposure to toxic chemical elements and developmental delays in children [7,8,13,27]. However, there is some inconsistency in findings, with some studies reporting no association between prenatal exposure and early child developmental impairment [28,29], while others find significant associations [30,31]. Some of the variability could be due to interaction and compounding effects.

The current study examined the effects of multi-chemical prenatal exposure on neurodevelopmental milestones for children aged 3–4 years born to mothers in the *Tanzania Mining and Health Cohort* in ASGM areas in northwestern Tanzania. The hypothesis is that Pb, Hg, As, and Cd will have synergistic or intensifying impacts on neurodevelopmental ability under joint exposure.

## 2. Materials and methods

### 2.1. Ethics statement

The joint Catholic University of Health and Allied Science and the Bugando Medical Center Ethics and Research Review Committee reviewed and approved the study protocol (CREC/038/2014, CREC/379/2019, and CREC/699/2023). Permission to conduct the research was obtained from Geita Regional and District Council local authorities. Written informed consent was obtained from mothers who participated in the study. For children below 5 years participation, individual mothers signed a special assent form permitting the assessment of their children neurodevelopmental status. Individual mothers had an opportunity to read and listen to the purpose, benefits, and risks of the research along with the rights of the participants before being asked for consent (*See* S1 Checklist). Individual children who were sick during the assessment were directed to seek medication at a nearby health facility and hence were excluded from the study. Mothers had an opportunity to know the developmental state of their children and benefited from the comprehensive counseling on early stimulation as a way to mitigate the impact of developmental delays.

### 2.2. Study design, study setting, and study population

This is a longitudinal follow-up study as part of the *Tanzania Mining and Health Cohort* in Geita District in northwestern Tanzania established in 2015 in ASGM areas. The details of this cohort have been explained elsewhere [26,32]; also see S2 Checklist. Briefly, the cohort included 883 pregnant women who were followed through to document the pregnancy outcomes. All children aged 3–4 years whose mother's chemical element levels were part of the *Tanzania Mining and Health Cohort* and still residing in the study area were recruited into the current study undertaken from 30th June 2019–25th May 2020. Mothers and their children in the cohort were excluded if they moved out of the ASGM area for over six months. A total of 352 mother-child pairs were recruited in this follow-up study, of which 310 were eligible, consented, and participated (Fig 1).

### 2.3. Data collection and statistical analysis

Parental data on socio-demographic and socio-economic characteristics of the caregivers (e.g., occupation, economic status, age, marital status, parity) and demographic characteristics of the child (e.g., sex, age, number of siblings in a family) were collected using a face-to-face structured interview questionnaire carried out at their homes. The economic status (SES) of the family was determined using a socioeconomic wealth quintile based on traditional indicators of wealth possession at the household level as detailed elsewhere [26,32], similar to those used in *Tanzanian Demographic and Health Surveys* [33]. Households that scored ≥9 on the SES assessment were classified as high-SES families; those that scored 6–9 were considered moderate SES families, while those <6 were considered low-SES families [25]. As co-variates in the regression analysis, children characteristics, including age, sex, parental education, and family socio-economic status, were descriptively analyzed. A forward-backward step-wise regression analysis was conducted to exclude all covariates

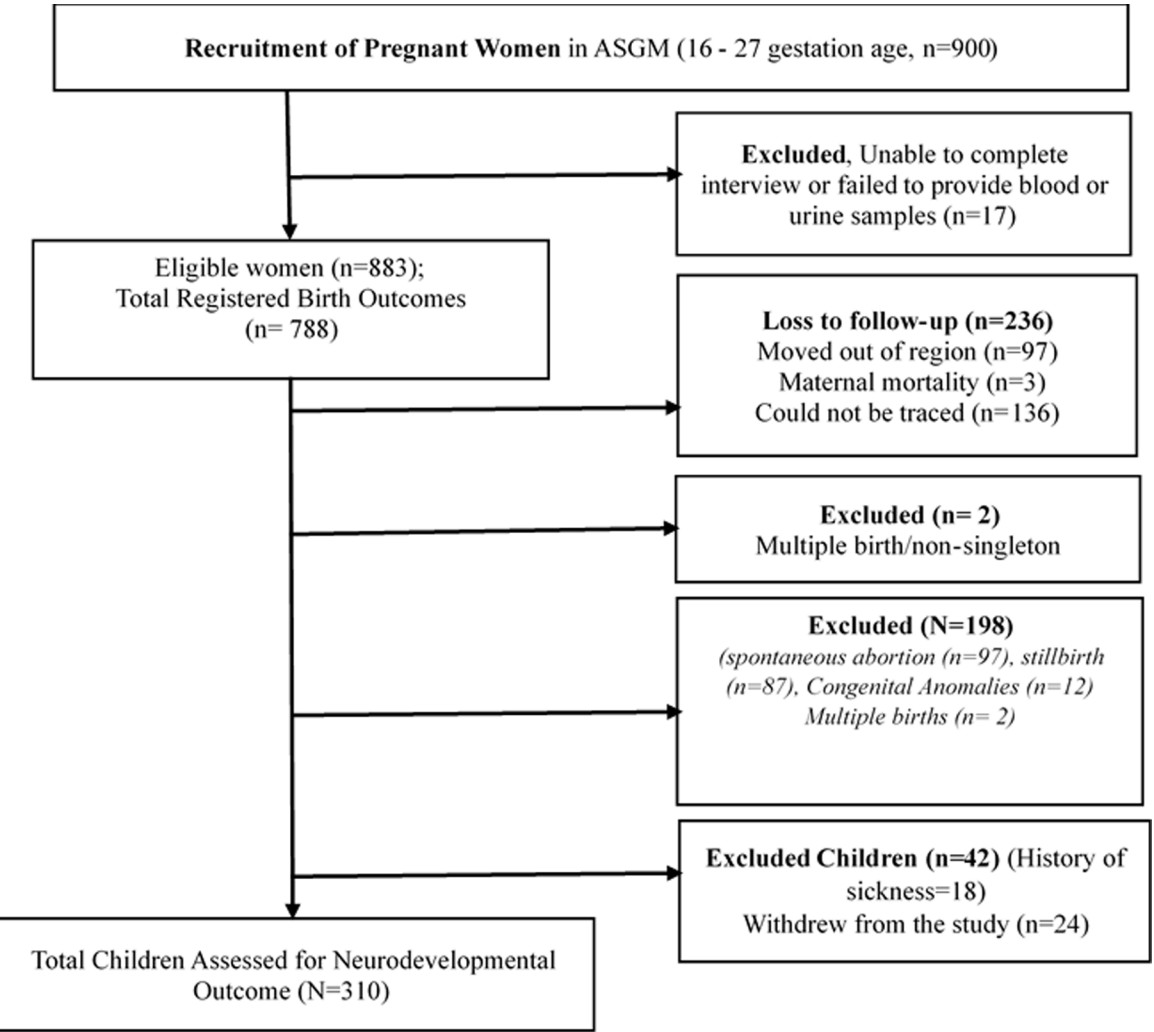

**Fig 1. Recruitment, exclusion, and eligibility, and loss to follow-up of mothers and children in ASGM areas.**

that did not improve model performance according to the Akaike Information Criterion (AIC) and Deviance using *Stata version 14.1* [*Stata Corp LP®*, College Station, Texas, USA) (*See* S1 and S2 Tables).

Data on mothers chemical element levels were utilized from the *Tanzania Mining and Health Cohort* [26,32] (*See* S1 Data). Chemical levels were estimated using dried blood spots (DBS) for Hg, Cd, and Pb and urine samples for As [25,31,33,34]. Spot urine samples were collected in acid-washed polyethylene containers preserved with 1% hydrochloric acid and stored below -8ºC before shipment to the laboratory. This was to prevent bacteria growth and absorption of the analytes. A drop of capillary blood was collected on filter paper following a simple finger prick for Cd, Pb, and Hg. Field and laboratory blanks (i.e., blank Whatman #903 filter papers) were included in sampling and analytical procedures to account for possible contamination and to avoid exposure misclassification. The DBS and urine samples underwent microwave-assisted closed vessel acid digestion to extract chemical element analytes from DBS and urine at 170ºC for 30 minutes. These analytes were determined by inductively coupled plasma mass spectrometry (ICP-MS; PerkinElmer, Shelton, CT, USA) (µg/L) at an ISO 17025 accredited laboratory (ALS Scandinavia, Sweden). Apart from prenatal exposure levels, postnatal exposure among the children enrolled in this study was not considered.

**Table 1. Parental Socio-demographic and socio-economic characteristics.**

| Parental Baseline characteristics | N | (%) |
|---|---|---|
| Maternal median age years (25, IQR: 20–30) | | |
| Maternal age group* | | |
| 14-24 | 107 | (34.5) |
| 25-34 | 148 | (47.7) |
| 35+ | 55 | (17.7) |
| *Mother education* | | |
| No formal | 61 | (19.7) |
| At least primary school | 249 | (80.3) |
| Social Economic Wealth Quintile (SEWQ) | | |
| Good or High (>9) | 211 | (68.1) |
| Moderate (6–9) | 64 | (20.6) |
| Low (<6) | 35 | (11.3) |
| *Mother's occupation* | | |
| Farm/Animal keeping | 244 | (78.7) |
| Mining | 9 | (2.9) |
| Business and public servant | 57 | (18.4) |
| *Father occupation* | | |
| Farm/Animal keeping | 98 | (31.6) |
| Mining | 144 | (46.5) |
| Business & Public Servant | 68 | (21.9) |
| *Marital Status* | | |
| Married | 285 | (91.9) |
| Single | 5 | (1.6) |
| Separated | 19 | (6.1) |
| Widow | 1 | (0.3) |
| *Relationship with partner* | | |
| Good and supportive | 284 | (91.6) |
| Poor (single-parent care) | 23 | (7.4) |
| Separated (child with grandparent) | 3 | (1) |
| *Individual Children Characteristic* | | |
| Mean age of children (43.9±3.4) months | | |
| *Sex of child* | | |
| Male | 163 | (52.7) |
| Female | 147 | (47.3) |
| *Presence of under-fives at home* | | |
| Yes | 346 | (78.8) |
| No | 93 | (21.2) |

Children's neurodevelopmental milestones were assessed using the *Malawi Developmental Assessment Tool* (MDAT), which has been developed and validated in low-income countries and is an appropriate tool for assessing children aged between 0–6 years [35]. The MDAT has been translated into Swahili the common language in Tanzania, validated in our local settings – and used in the *Tanzania Mining and Health Cohort* [8]. The MDAT contains 136 items, with 34 items across four domains (gross motor, fine motor, language, and social). The neurodevelopmental assessment was conducted by trained public health nurses blinded to the children's prenatal exposure concentrations and [8] was determined

**Table 2. Maternal prenatal blood total mercury (Hg), blood total cadmium (Cd), blood total lead (Pb), and total urinary arsenic (As) (N = 310).**

| Prenatal exposure levels | N | (%) | P10 | P25 | Median (95%CI) | P75 | P95 | Max |
|---|---|---|---|---|---|---|---|---|
| Total As (µg/L) | | | 2.8 | 4.5 | 8.3 (7.4–9.3) | 14.9 | 38.2 | 213 |
| ≥ 15 | 76 | (24.6) | | | – | | | |
| <15 | 234 | (75.4) | | | – | | | |
| Total Mercury (µg/L) | | | 0.6 | 0.8 | 1.2 (1.1–1.3) | 1.7 | 3.8 | 72 |
| ≥0.8 | 237 | (76.5) | | | – | | | |
| <0.8 | 73 | (23.4) | | | – | | | |
| Total Cadmium (µg/L) | | | 0.14 | 0.21 | 0.2 (0.2–0.21) | 0.3 | 0.5 | 1.5 |
| ≥0.3 | 49 | (15.7) | | | | | | |
| <0.3 | 261 | (84.3) | | | | | | |
| Total Lead (µg/L) | | | 11.6 | 17.2 | 27.2 (25–30.3) | 42.5 | 85.8 | 145.1 |
| <35 | 194 | (62.6) | | | | | | |
| ≥35 | 116 | (37.4) | | | | | | |

**Note:** We dichotomized Pb, Hg, Cd, and As based on human biomonitoring reference values established by the GerESIV of 35.0 µg/L, 0.80 µg/L, 0.30 µg/L for blood lead, mercury, and cadmium, and 15 µg/L for urinary arsenic (Schulz et al. 2009). CI, confidence interval; ug/l, P, percentile; ug/l, microgram/liter.

based on the total score (ranging from 0 to 34 for each domain) [7]. Children who performed ≥ 90th percentile on all of the items in that domain or <90th percentile on one or two items in that domain were classified as usual [35]. Those who performed <90th percentile on more than two items in a domain were classified as impaired. In addition, individual children were classified in terms of global neurodevelopment status. In contrast, those who scored normal outcomes on all domains were classified as typically developing, and those impaired on at least one domain were classified as impaired for neurodevelopmental outcomes [7]. In the present study, the median motor scores were 11 (IQR = 9–13), of which the minimum was 4 and the maximum was 15, while the alpha coefficient for the items was 0.89. The median language scores were 7 (IQR = 6–7), with a minimum score of 4 and a maximum of 20, while the alpha coefficient for the items was 0.91. The median social functioning score was 10 (IQR = 8–11), of which the minimum was 5, and the maximum was 19, with an alpha coefficient of 0.94. In all domains, the alpha coefficient suggests good internal consistency.

Assessing the joint effects of multiple toxic chemical exposures on childrendevelopmental milestones is challenging with traditional regression models because: 1) they are unable to account for internal correlation/multicollinearity between chemical elements, and 2) they are unable to optimize each chemical weight or contribution on the overall joint effect. As a result, we used the quantile g-computation approach. Even though weighted quantile sum (WQS) regression has been used to make causal inferences for multi-chemical exposures on health outcomes, forcing chemical weights to have unidirectional effects (positive direction) in the WQS has been critiqued for introducing bias and inconsistency in model parameters. Therefore, proposed a more flexible and robust g-computation approach to address previously identified challenges in joint exposure modelling with the WQS model has been recommended [36]. For the dose-response relationship of joint exposure using quantile g-computation, parameters are estimated as:

$$Y_i = \beta_0 + \psi \sum_{j=1}^{d} w_j X_{ji}^q + \varepsilon_i = \beta_0 + \psi \sum_{j=1}^{d} \beta_j X_{ji}^q + \varepsilon_i \tag{1}$$

**Where:** $X_{ji}^q$ is the quantized version of exposure $X_j$, $\psi$ is the weight given as $\sum_{j=1}^{d} \beta_j$ (where $\beta_j$ is the effect size of exposure $j$).

The sum of all weights $\psi = 1$; therefore, the individual weight for each heavy metal is given as:

$$w_k = \frac{\beta_k}{\sum_j^d \beta_j} \tag{2}$$

PLOS Global Public Health

**Table 3. Unadjusted and covariate-adjusted model on chemical elements mixture effects on neurodevelopmental abilities.**

**Unadjusted Model**

| | General Impairment cPR (95% Lower CI, Upper CI) | Social Behavior cPR (95% Lower CI, Upper CI) | Language Ability cPR (95% Lower CI, Upper CI) | Gross Motor cPR (95% Lower CI, Upper CI) | Fine Motor cPR (95% Lower CI, Upper CI) |
|---|---|---|---|---|---|
| Intercept | 0.646 (0.522, 0.800) | 0.118 (0.0542, 0.2571) | 0.504 (0.352, −0.324) | 0.644 (0.519, 0.799) | 0.4454 (0.3137, 0.6325) |
| psi1 | 0.8811 (0.759, 1.023)† | 1.149 (0.728, 1.815) | 0.418 (0.302, 0.579)*** | 0.811 (0.6843, 0.961)* | 0.943 (0.7540, 1.1801) |
| Cd+As | 1.172 | 1.642 (Cd+Pb+As) | 0.307 (Pb+Hg+Cd) | 0.612 (Pb+Hg+As) | 0.763 (Pb+Hg+As) |
| Pb+Hg | 0.645 | 0.717 (Mercury) | 1.009 (Arsenic) | 1.074 (Cd) | 1.188 (Cd) |
| Pb weight | 0.686 (−) | 0.350 (+) | 0.632 (−) | 0.658 (−) | 0.509 (−) |
| As weight | 0.275 (−) | 0.027 (+) | 1 (+) | 0.330 (−) | 0.0321 (−) |
| Cd weight | 0.725 (+) | 0.623 (+) | 0.143 (−) | 0.012 (−) | 1 (+) |
| Hg weight | 0.314 (−) | 1 (−) | 0.225 (−) | 1 (+) | 0.459 (−) |

**Covariate Adjusted Model**

| | General Impairment aPR (95% Lower CI, Upper CI) | Social Behavior aPR (95% Lower CI, Upper CI) | Language Ability aPR (95% Lower CI, Upper CI) | Gross Motor aPR (95% Lower CI, Upper CI) | Fine Motor aPR (95% Lower CI, Upper CI) |
|---|---|---|---|---|---|
| (Intercept) | 0.659 (0.545, −0.225) | 0.132 (0.0635, 0.2752) | 0.463 (0.313, 0.684) | 0.633 (0.512, 0.784) | 0.459 (0.329, 0.640) |
| psi1 | 0.866 (0.747, 1.01)* | 1.067 (0.694, 1.641) | 0.446 (0.313, 0.636)*** | 0.822 (0.699, 0.966)* | 0.923 (0.739, 1.153) |
| Pb+As+Hg | 0.632 | 1.586 (Pb+Cd) | 0.3296 (all) | 0.600 | 0.735 |
| Cd | 1.125 | 0.682 (Hg+As) | ---------- | 1.095 | 1.182 |
| Pb weight | 0.593 (−) | 0.395 (+) | 0.610 (−) | 0.499 (−) | 0.459 (−) |
| As weight | 0.036 (−) | 0.002 (−) | 0.036 (−) | 0.142 (−) | 0.118 (−) |
| Cd weight | 1 (+) | 0.605 (+) | 0.131 (−) | 1 (+) | 1 (+) |
| Hg weights | 0.371 (−) | 0.998 (−) | 0.223 (−) | 0.359 (−) | 0.423 (−) |

**Note:**

†=significant at 0.1;

*=significant at 0.05;

**=significant at 0.001;

***=significant at 0.0001; psi1=combined effects of all chemical mixers; neg=negative weight; pos=positive weight; CI=confidence interval; aPR=adjusted prevalence ratio; cPR=crude Prevalence ratio.

**Where:** $w_k$ is the weight for each chemical element, whereby weights can take on negative or positive directions based on the dose-response pattern between the health outcome and the exposure under consideration. Since our dependent variable was binary for each of the development components, we estimated the model $Pr(Y = 1 | X^d)$ with logistic regression and then converted to log (RR) (calculated automatically by the *g-comp* package in *RStudio*).

Three separate models were built: (1) the multi-chemical model, where all the chemicals were entered but unadjusted for confounding; (2) the multi-chemical model, where all the chemicals were entered and adjusted for confounding; and (3) a chemical-specific model; where one chemical was entered at a time and adjusted for confounding effect. An antilogarithm of the log (RR) was used to convert it to a prevalence ratio (PR) for straightforward interpretation. We calculated 95% confidence intervals (CIs) around point estimates with 1000 bootstrap resampling iterations with Monte-Carlo simulations to address uncertainties introduced from the sample size effect. Relationships were significant at a 5% level. All modelling exercises were conducted with the *qcomp* package in *RStudio version 4.3.2* [37].

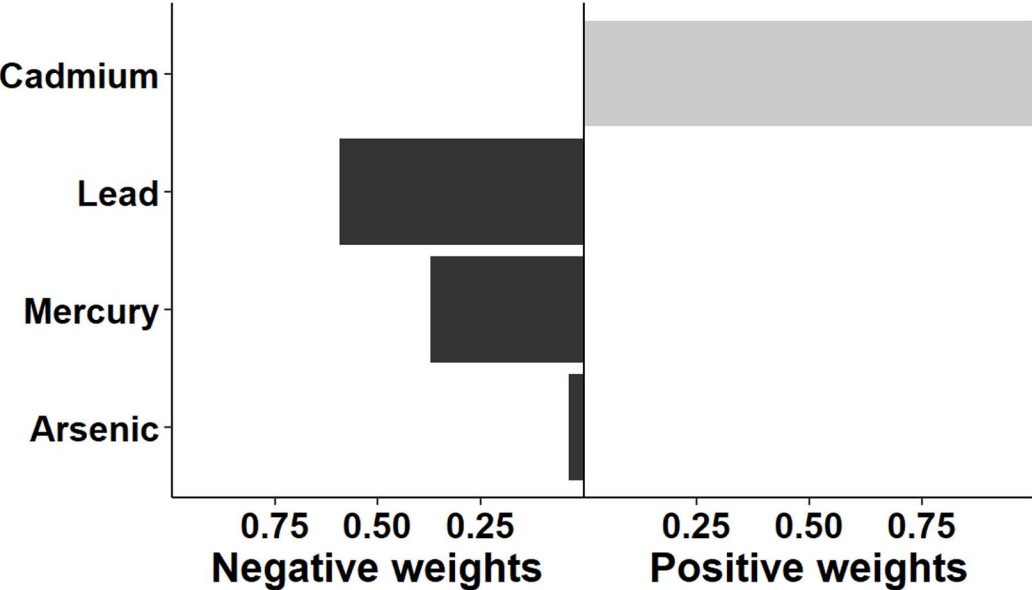

**Fig 2. Effects of individual chemical elements and their weights on neurodevelopmental impairments in mixed interaction models for (a) Global Neurodevelopmental status.** The bar plots pointing to the right represent chemicals that contributed positively (increased risk) to the joint exposure risk. In contrast, those pointing to the left indicate chemicals inversely contributing to the joint exposure risk.

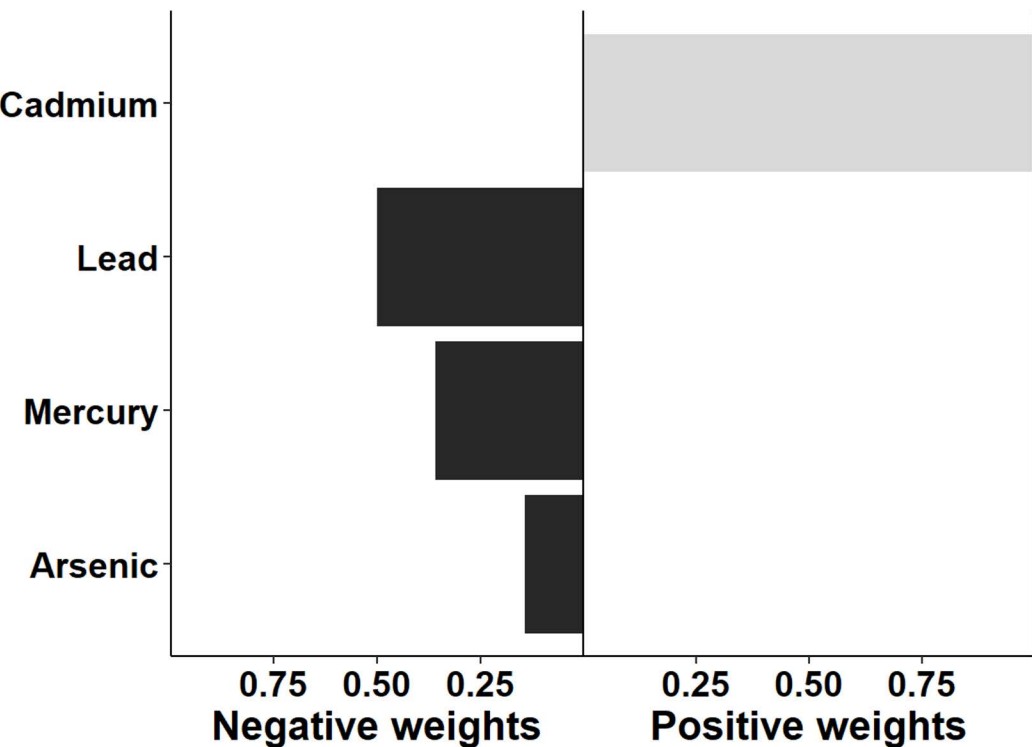

**Fig 3. Effects of individual chemical elements and their weights on neurodevelopmental impairments in mixed interaction models for (b) Gross Motor.** The bar plots pointing to the right represent chemicals that contributed positively (increased risk) to the joint exposure risk. In contrast, those pointing to the left indicate chemicals inversely contributing to the joint exposure risk.

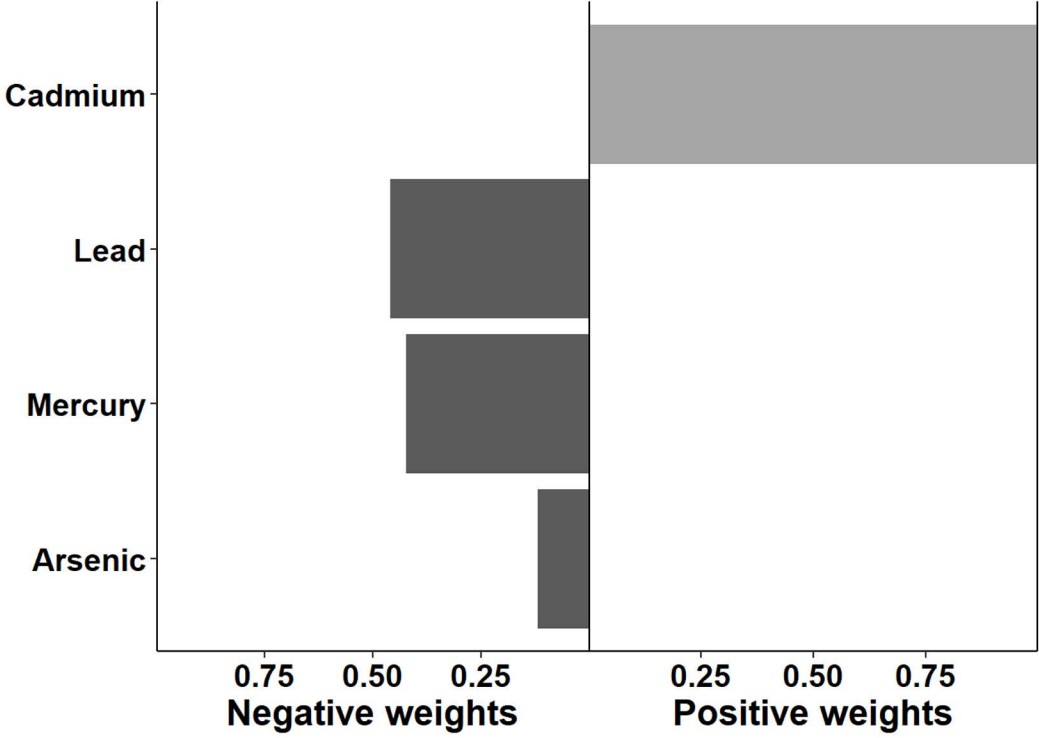

**Fig 4. Effects of individual chemical elements and their weights on neurodevelopmental impairments in mixed interaction models for (c). Fine Motor.** The bar plots pointing to the right represent chemicals that contributed positively (increased risk) to the joint exposure risk. In contrast, those pointing to the left indicate chemicals inversely contributing to the joint exposure risk.

## 3. Results

Table 1 summarizes the characteristics of parents and children. The median age of mothers was 25 (IQR = 20–30), and 69.7% were between 20 and 34 years. Most mothers (80.3%; n = 249) had at least a primary school education. Based on asset ownership assessment, 91% (n = 282) had moderate and high SES. Most of the mothers (n = 244; 78.7%) were either farming or keeping animals and only nine mothers (2.9%) were in mining. Among fathers, more than a quarter (n = 144; 46.5%), were working in mining. The average age of children was 3.7 (SD = 0.3) years and more than half, 225 (51.4%) were females.

Table 2 summarizes maternal prenatal chemical element levels from the initial study of the *Tanzania Mining and Health Cohort* in the current study. The total median urinary As level was 8.3 (IQR = 4.5-14.9) µg/L; more than three-thirds (n = 234; 75.4%) of mothers had less than 15µg/L of urinary As (Table 3). The median total level of blood Hg was 1.2 µg/L (IQR = 0.8-1.9) among mothers, and the majority (n = 237; 76.5%) had higher than ≥0.8 µg/L. The median prenatal Cd levels were 0.20 (IQR = 0.2-0.3) µg/L, with the majority (n = 261; 84.3%) of mothers having Cd levels <0.30 µg/L. Mothers' median prenatal blood Pb levels were 27.2 µg/L (IQR = 17.2-42.5). Over one-third of children were prenatally exposed to Pb levels above 35 µg/L.

The prevalence of neurodevelopmental impairment among the study participants was above 15% across all domains. Specifically, 20.3%, 23.9%, 28.3, and 16.2% of the participating children were impaired in their neurodevelopmental status for gross motor, fine motor, language ability, and social interaction ability, respectively. Additionally, for global

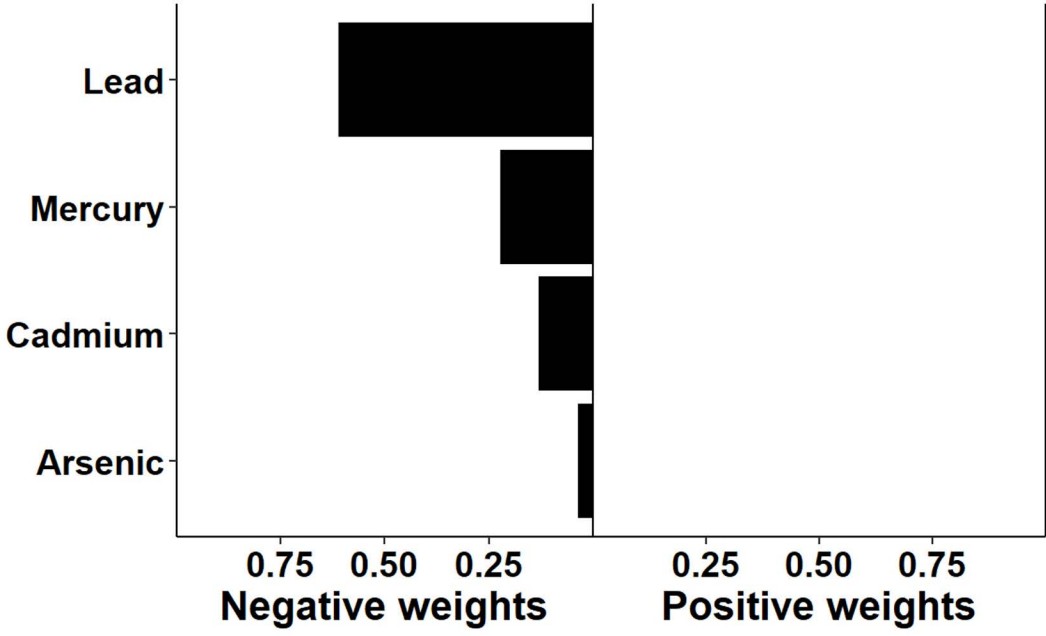

**Fig 5. Effects of individual chemical elements and their weights on neurodevelopmental impairments in mixed interaction models for (d) Language Ability.** The bar plots pointing to the right represent chemicals that contributed positively (increased risk) to the joint exposure risk. In contrast, those pointing to the left indicate chemicals inversely contributing to the joint exposure risk.

neurodevelopmental status, slightly above half (50.6%) of the participating children were impaired in at least one of the neurodevelopmental domains.

The association between multi-chemical exposures and the five domains of cognitive developmental outcomes (fine motor, gross motor, social functioning, language ability, and general neurodevelopmental impairment) are reviewed in the following sections. Figs 2–11 below show the joint exposure effects for the five domains.

One quantile increases in the Pb, Hg, Cd, and As joint exposure reduced the global neurodevelopmental performance by 13.34% (aPR = 0.866; 95% CI: 0.747,1.005; $p$ = 0.047) (Table 3). In a chemical-specific model, Pb reduced global neurodevelopmental performance by 10.5% (aPR = 0.895, 95% CI: 0.810- 0.989; $p$ = 0.030). However, in a mixed-effect model, the Pb level significantly decreased global neurodevelopmental performance by 23.8% (aPR = 0.762; $p$ = 0.02) (Table 4).

In the chemical-specific model, Hg reduced global neurodevelopmental performance by 7% (aPR = 0.930, 95% CI: 0.853-1.014; $p$ = 0.10) (Table 5), whereas in a mixed-model, Hg (aPR = 0.8446) decreased it by 15.6. Furthermore, compared to boys, girls had a 39.3% (aPR = 0.607; $p$ = 0.040) decline in global neurodevelopmental performance. Also, a unit change in age was statistically significantly associated with a 6.1% increase in global neurodevelopmental performance (aPR = 1.061; $p$ = 0.015). The mother's level of education had no protective effect, even though, compared to those without formal education, those with ordinary education levels (aPR = 0.308; $p$ = 0.017) had a nearly 71% decline in global neurodevelopmental performance.

Pb, Hg, Cd, and As jointly reduced gross motor ability by 17.8% (aPR = 0.822; 95% CI: 0.699, 0.966; p = 0.041) (Table 3). In a chemical-specific model, Pb reduced gross motor ability by 11.5% (aPR = 0.885, 95% CI: 0.793- 0.988; $p$ = 0.030), while in a mixed model, Pb decreased gross motor by 22.5% (aPR = 0.7752, $p$ = 0.028) (Table 4). Similarly, Hg reduced gross motor ability by 8.8% in a chemical-specific model (aPR = 0.912, 95% CI: 0.824- 1.010; $p$ = 0.078) (Table 5), and 16.8% (aPR = 0.832; $p$ = 0.094) in the mixed model. As reduced gross motor ability by 4.8% (aPR = 0.952, 95%

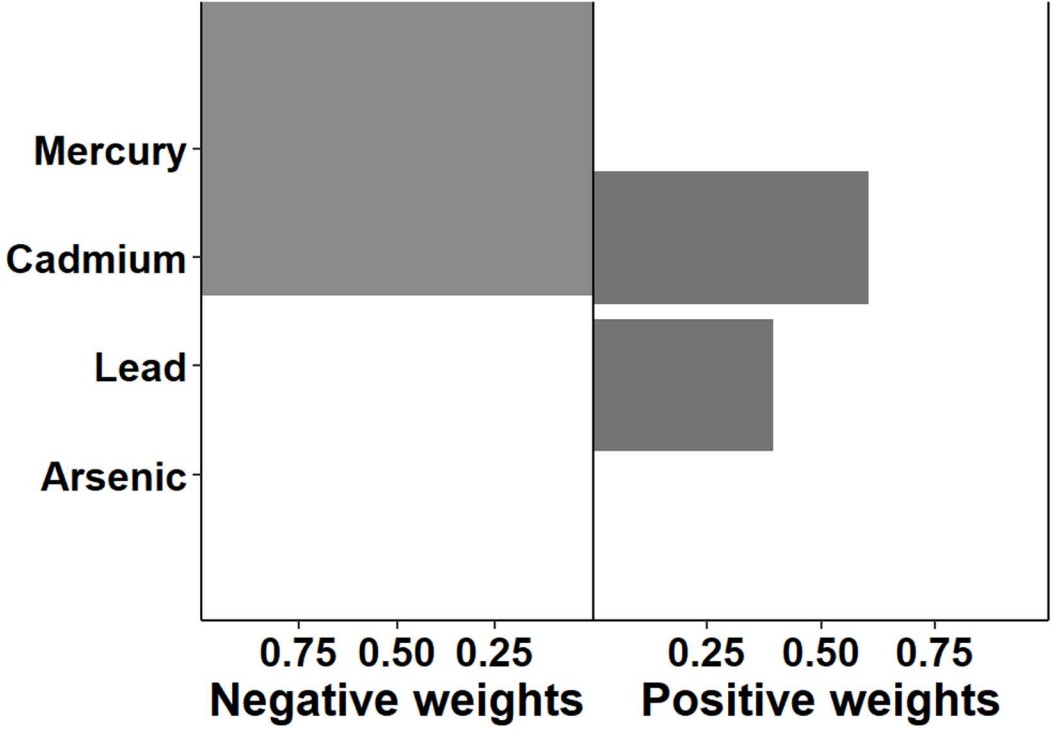

**Fig 6. Effects of individual chemical elements and their weights on neurodevelopmental impairments in mixed interaction models for (e) Socio-interactions.** The bar plots pointing to the right represent chemicals that contributed positively (increased risk) to the joint exposure risk. In contrast, those pointing to the left indicate chemicals inversely contributing to the joint exposure risk.

CI: 0.858-1.056; $p = 0.354$) in single-element exposure, while in the mixed model, it reduced gross motor ability by 7% (aPR = 0.930; $p = 0.518$). Additionally, children born to parents with ordinary education levels decreased their gross motor performance by 79.83% (aPR = 0.201; $p = 0.002$). Furthermore, age was statistically significantly related to gross motor ability (aPR = 1.054; $p = 0.035$).

Although not statistically significant, Pb, Hg, and As levels jointly negatively affected fine motor performance (aPR = 0.763), while Cd produced a positive effect (aPR = 1.188) (Table 3). Even though no association was found, Pb still made the most significant contribution as a risk factor (45.9%), followed by Hg (42.3%) and As (11.8%). Children from low SES backgrounds had a 62.7% decrease in fine motor performance compared to higher SES (aPR = 0.373; $p = 0.015$) (Table 4).

The combined effects of all the chemical elements decreased language ability by 55.4% (aPR = 0.446; 95% CI: 0.313, 0.636; $p < 0.001$) (Table 3). In a chemical-specific model, Pb reduced language ability by 43.9% (aPR = 0.561, 95% CI: 0.434-0.726; $p < 0.001$) while in a mixed model, it decreased language ability by 49.1% (aPR = 0.509; $p < 0.0001$) (Table 4). Furthermore, individually, Hg reduced language ability by 22.8% (aPR = 0.772, 95% CI: 0.621-0.959; $p = 0.019$) (Table 5), while in the mixed model, Hg decreased language ability by 21.9% (aPR = 0.781; $p = 0.084$). Cd (aPR = 0.777, 95% CI: 0.620- 0.973; $p = 0.028$) significantly reduced language ability by 22.3%, but the association was insignificant in a mixed model (Cd: aPR = 0.865; $p = 0.344$). Additionally, sex (aPR = 0.926; $p = 0.810$), mother's primary school education level (aPR = 0.837; $p = 0.645$), the child's age (aPR = 1.0519; $p = 0.185$), moderate SES (aPR = 1.711; $p = 0.141$), and low SES (aPR = 1.345; $p = 0.520$) were not statistically significantly related to language ability.

Although not statistically significant, the joint effect of all the chemical elements was positively associated with social functioning at 6.7% (aPR = 1.067; 95% CI: 0.694, 1.641; p > 0.10) (Table 3). While the combination of Pb and Cd had a positive joint impact

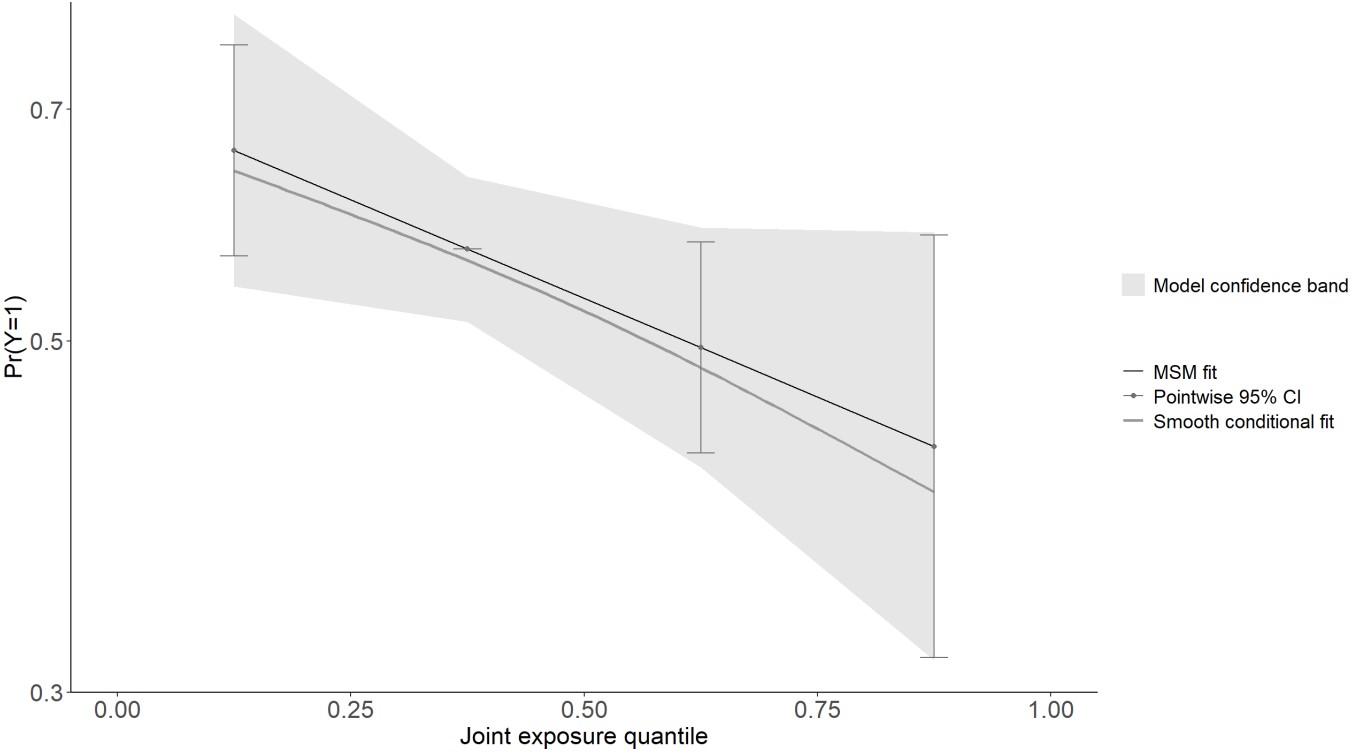

**Fig 7. I of the model uncertainty band for (a) Global Neurodevelopmental status.** The y-axis indicates the probability of not being cognitively impaired, while 4 the x-axis shows the quantile distribution of the joint exposures. The MSM fit is an estimate 5 from the marginal structural model.

(aPR = 1.585), Hg and As provided a joint antagonistic effect (aPR = 0.682), thus nullifying the overall combined effect. However, in the negative direction, prenatal exposure to Hg had the highest weight (99.78%), followed by As (0.225%), while in the positive direction, Cd had the most significant weight (60.5%), followed by Pb (39.5%). Even though no association was observed for either the chemical-specific or mixed model, Hg had a potentiation effect, while other elements remained relatively unchanged.

## 4. Discussion

Fifty percent of the children in this current study had neurodevelopmental impairment in one or more domains, which is higher than when these children were assessed (31%) in infancy [8]. Children developed additional impairments at three to four years, providing evidence that neurodevelopmental deficits from prenatal exposures continue later in life [31]. The increasing severity and prevalence of neurodevelopmental impairment among children as they age could occur for a variety of reasons. Firstly, brain development begins earlier in embryonic life and continues well beyond childhood into adulthood [38,39]. During these early stages, the brain cells undergo mitosis and migrate, differentiate, develop synaptic connections, and undergo programmed apoptosis [39]. Therefore, any interference from toxic exposures may alter subsequent developmental milestones [38,40]. Even a slight disruption from Pb and Hg at a fetal or infancy stage may continue to have long-term cascade effects later in life, including learning deficits [41]. Secondly, prenatally exposed children may continue to be exposed to even higher environmental pollutants after birth. Exposure at the community level also occurs from home, play, and travel, including to-from and at school [42–44]. Drinking contaminated water or eating soil from the ground may add to the existing health burden carried by the prenatal exposure stages [44].

Generally, Pb and Hg appeared to induce a toxic effect on neurodevelopmental abilities among children. Significantly, when interacting with other chemical elements, their individual marginal effects are amplified beyond what would be

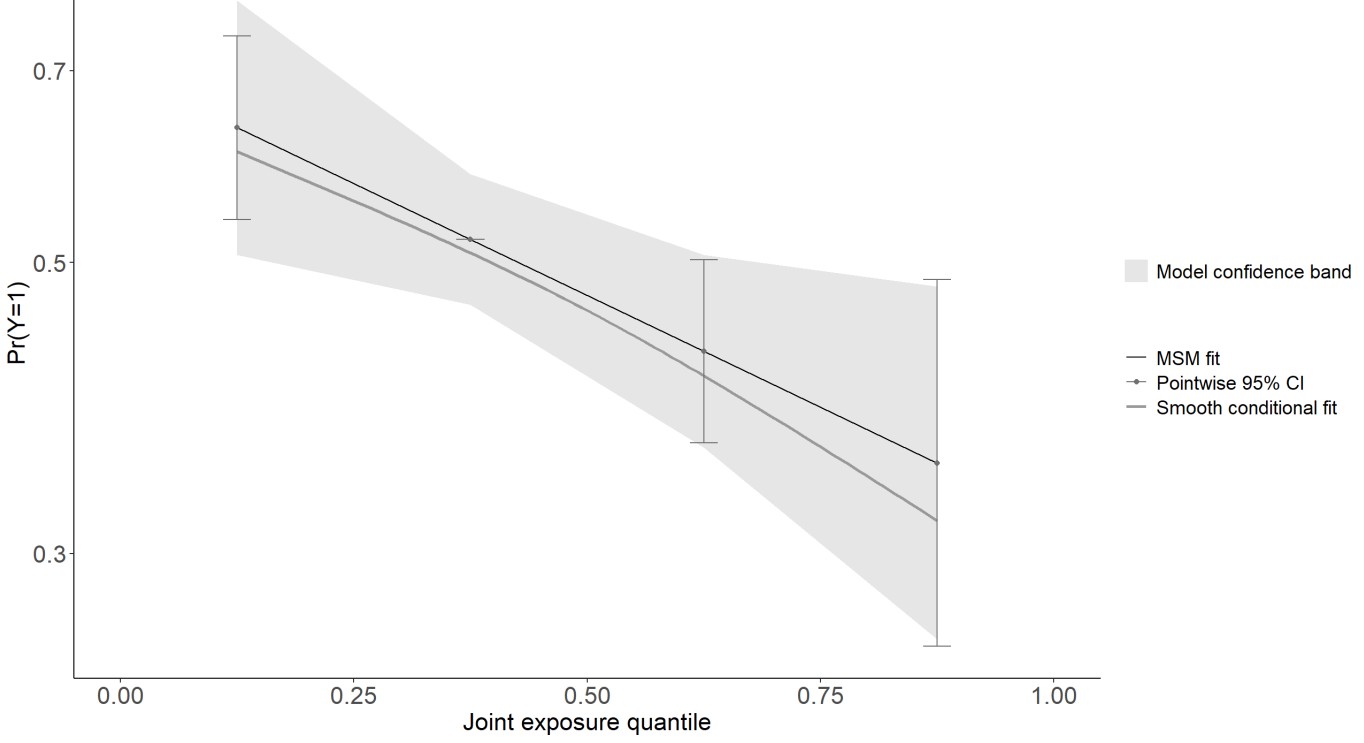

**Fig 8. l of the model uncertainty band for (b) Gross Motor.** The y-axis indicates the probability of not being cognitively impaired, while 4 the x-axis shows the quantile distribution of the joint exposures. The MSM fit is an estimate 5 from the marginal structural model.

expected under the additive assumption. Given the magnitude of neurodevelopmental impairment may increase as the child grows and even worsens when the child is multi-chemically exposed, stringent environmental control measures that limit toxic exposures are needed. Further, setting new safety reference levels that integrate multi-chemical exposure scenarios is necessary.

Joint exposure to Pb, As, Hg, and Cd increased the risk of neurodevelopmental impairment in one or more domains (the global neurodevelopmental impairment). While blood Pb and Hg were statistically significantly related to global neurodevelopmental impairment, Pb carried the most significant weight in the mixed model (contributing to about 59.3% of the total weight), suggesting its dominant role in inducing neurodevelopmental impairment. Compared to those with no formal education, children whose mothers had ordinary-level education had about a 70% reduced global neurodevelopmental ability. This trend suggests the need to consider how sociodemographic disparities and gender differences interact with multi-chemical exposures to elevate neurodevelopmental impairment among children.

Combined Pb, As, Cd, and Hg exposure was linked to decreased gross motor ability. Accounting for confounding effects, Pb independently reduced gross motor. In the presence of other chemicals, the negative impact of Pb on gross motor ability was higher than expected under the additive assumption. Our findings corroborate other international studies. For instance, prenatal exposure to Pb in the third trimester significantly decreased the development index of infants at 6 months of age [5]. In a Tanzanian study, Pb and As independently did not induce neurodevelopmental impairment among infants. However, when they acted in the presence of Hg, there was a potentiated outcome [8]. This may be explained by the neurotoxic effect of Pb on nerve cells [45]. Cell damage in the hippocampal and volt sections of the brain due to Pb exposure may affect execution function, including a decrease in gross motor function. Further, since Hg, Cd, and As may have similar biokinetic pathways for inducing toxicity, the marginal effect of Pb may be amplified when exposed jointly with

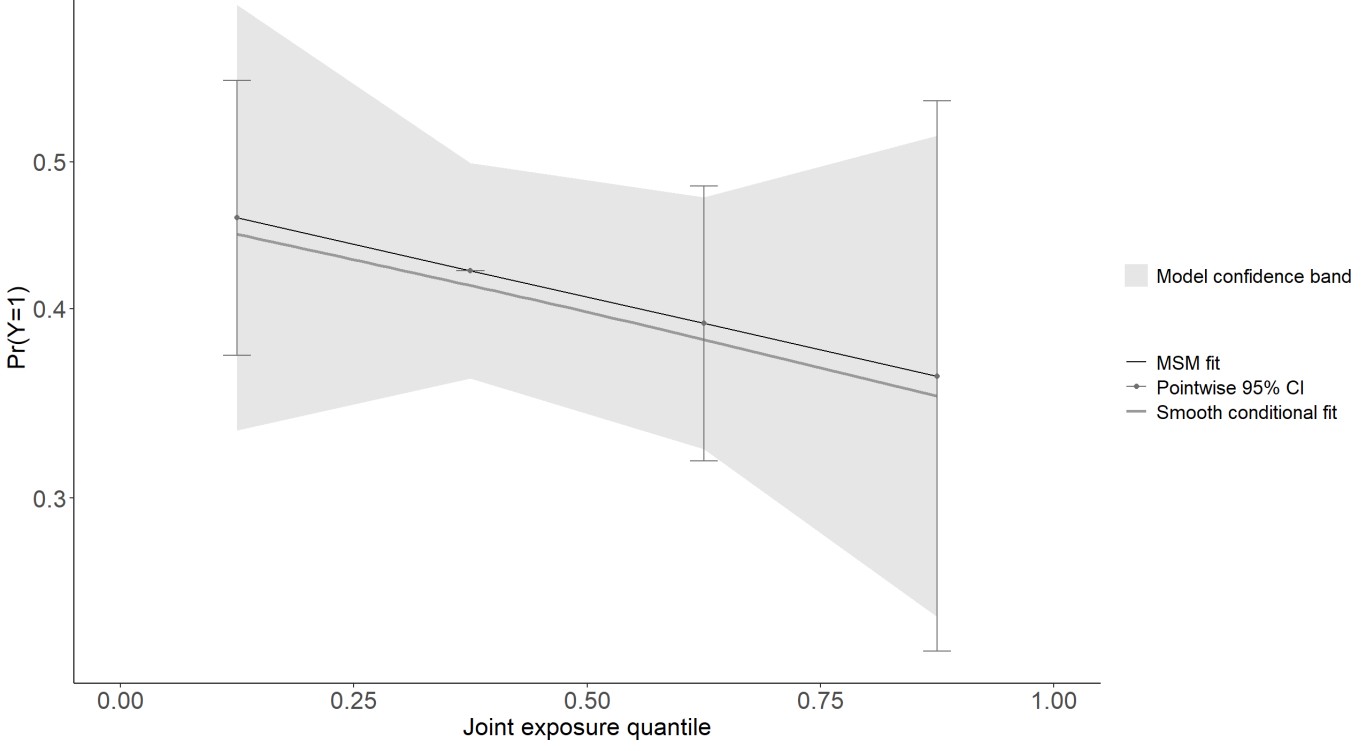

**Fig 9. I of the model uncertainty band for (c) Fine Motor.** The y-axis indicates the probability of not being cognitively impaired, while 4 the x-axis shows the quantile distribution of the joint exposures. The MSM fit is an estimate 5 from the marginal structural model.

other heavy metals. Clinical evidence suggests that the combined impact of Pb, Cd, and As can elevate neural oxidative stress by increasing the levels of $Ca^{2+}$ [46]. They can also jointly stimulate extracellular signal-regulated kinase (ERK), c-Jun N-terminal kinases (JNK), and mitogen-activated protein kinase3 (MAPK3) pathways, which may cumulatively affect various neurological and psychomotor functions in children [46]. Even though Pb independently reduced the mental development index as reported among South Korean children, the effect amplified when it interacted with Hg and Cd [23]. A similar synergistic effect of Pb in the presence of Hg and polychlorinated biphenyls has been found between neurophysiological disorders and Pb exposure among children in Montreal, Canada [24].

Elevated Hg levels were linked to decreased gross motor function, accounting for approximately 36% of the total risk weight. Even though Hg independently reduced gross motor, the impact was higher in the presence of other metallic elements, suggesting a synergistic effect. For example, a quartile increase in Hg was linked to an 8.78% decrease in gross motor ability and about 16.8% when Hg interacted with Pb, Cd, and As. In a previous study from the US, poor psychomotor development among children at 6 months was associated with elevated prenatal Hg exposure [47]. The synergistic effect of Hg in the presence of Pb and Cd is equally reported in a study investigating cognitive development and multi-chemical exposures in South Korea [23]. Methylmercury, Pb, and As have similar neurological pathways, including disrupting the expression of N-methyl-D-aspartate (NMDA) receptors [48]. Therefore, their individual effects may be more than additive when exposed in combination. We did not, however, find any association between Cd and As and gross motor ability. This may be explained by the restrictive passage of Cd through the placenta from the protein metallothionein [49,50]. Nevertheless, we found that their impact was elevated in the presence of Hg and Pb, suggesting they may also have potentiating effects. More studies are therefore needed to provide concrete evidence better.

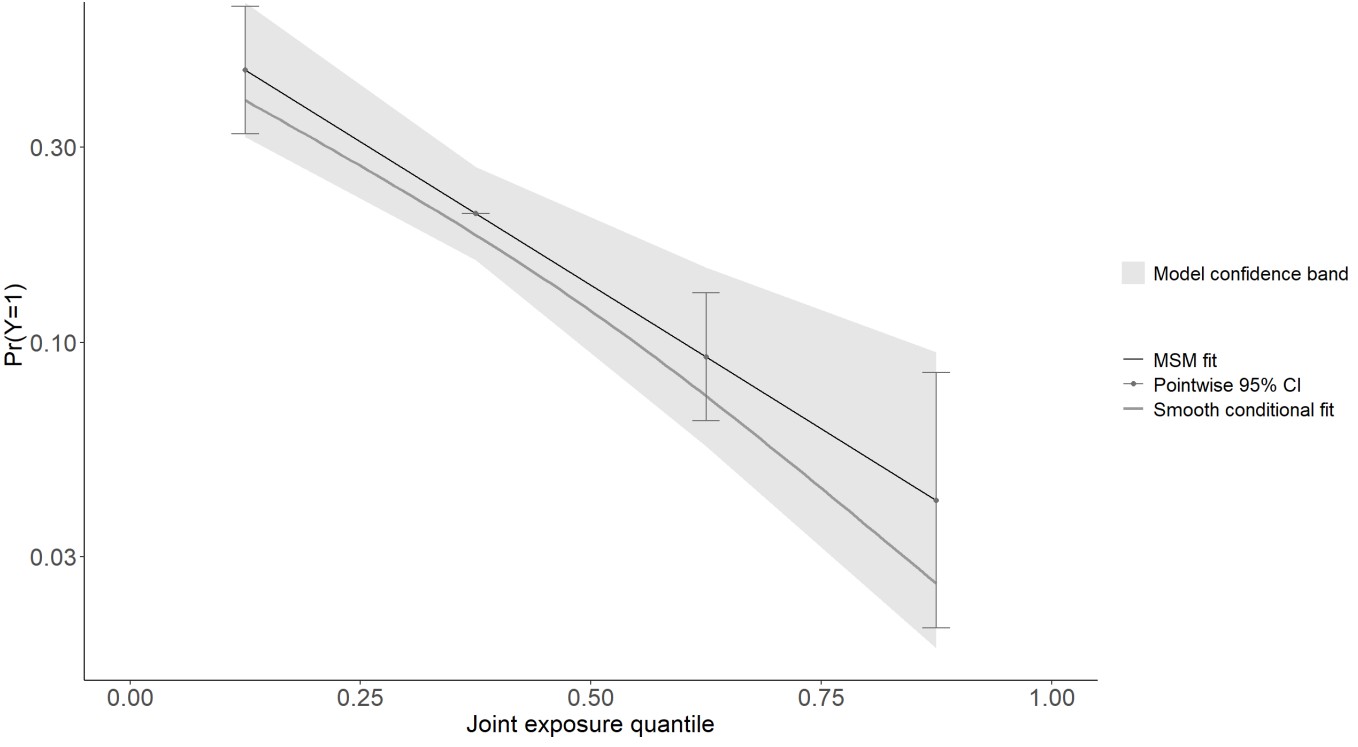

**Fig 10. I of the model uncertainty band for (d) Language Ability.** The y-axis indicates the probability of not being cognitively impaired, while 4 the x-axis shows the quantile distribution of the joint exposures. The MSM fit is an estimate 5 from the marginal structural model.

Although the association was not statistically significant, Pb, Hg, As, and Cd reduced fine motor ability in multi-chemical and single-element models. Similarly, no association was found between heavy metals exposure and social interaction in single and multiple-chemical models. In the social interaction model, besides Hg, which was inversely linked, we observed a positive association with other heavy metals. This effect direction contradicts other studies' reported effect direction [46,51,52], although it is possible the association in the current study was due to random chance.

The combined effect of Pb, As, Hg, and Cd was statistically significantly associated with decreased language ability, with Pb playing a significant role in the joint effect. Independently, the impact of Pb was synergistic, as the risk of language ability decline increased from 43% in single-element exposure to 49% in multi-chemical exposure. In addition to the reductive effect of Pb exposure on IQ scores, memory, social interaction, and deficits in language ability are observed among children exposed to Pb contamination [46]. Exposure to Pb is highly damaging to the central nervous system (CNS) [46]. Pb exposure affects the CNS's dopaminergic, cholinergic, and glutamatergic systems. For instance, an abnormal release of dopamine transporters is linked to Pb exposure, which can increase language and communication difficulties [46,53]. Additionally, the cholinergic system known to regulate energy metabolism can be affected by Pb exposure [46,54].

In a recent study, children who resided in Pb-contaminated neighborhoods had smaller volumes of the mid-anterior and mid-posterior corpus callosum [55]. The decline in these systems affected processing and language speed [55]. In a large *National Health and Nutrition Examination Survey (NHANES)* study, about 1,025,695 people in the U.S. who were born between 1989 and 1998 and had difficulties in listening, reading, reasoning, or speaking were exposed to elevated Pb levels [56]. Similarly, exposure to > 22.2ppm of Pb was linked to an 8.6-point decrease in word identification test compared to those exposed to 5.99ppm [46]. As the primary target may involve the CNS, the health consequences may be long-term extending from infancy to adulthood. This may explain why about 50% of children in our current study had

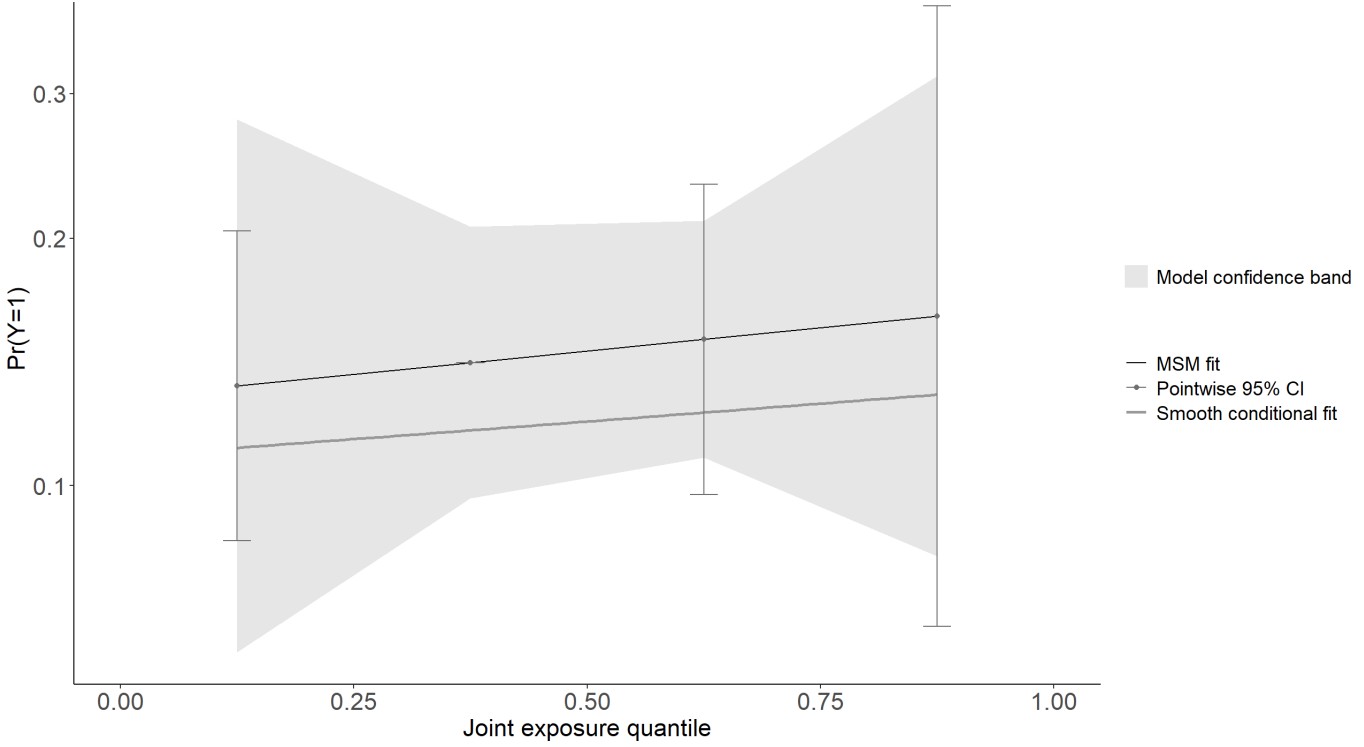

**Fig 11. I of the model uncertainty band for (e) Socio-interactions.** The y-axis indicates the probability of not being cognitively impaired, while 4 the x-axis shows the quantile distribution of the joint exposures. The MSM fit is an estimate 5 from the marginal structural model.

neurodevelopmental impairment in at least one of the four domains, compared to 31% during infancy. The effects may worsen as the child grows and gets exposed to multiple pollutants from diverse sources.

Hg exposure significantly reduced language ability in single-element and multi-chemical exposure models. Children with an increased level of Hg in their hair, fingernails, and toenails are more likely to score lower in the Bayley-III scale score of expressive language [57]. Other studies support this evidence [58–60]. Our results also support a study in Spain where prenatal Hg exposure has been associated with poor performance in verbal function at 4–5 years of age [59]. However, in the reactive presence of other heavy metals, a decreased risk of Hg exposure, from 54% risk to 22% was discovered. This may be due to different elements' antagonistic effects in the joint model. Even though a quartile increases in Cd and As were associated with a 4% and 14% decrease in language ability, respectively, they were not statistically significant. This is supported by the finding in the Korean population study among children, where early pre-natal Cd exposure had no significant association with cognitive and psychomotor developmental impairment, even with increased Pb levels at 6 months of age [61]. Similar findings have been reported in the US *National Health and Nutri-tion Examination Survey* (NHANES) [62]. Since this study considered a marginal population-level effect, we recom-mend that future studies focus on sub-group studies to better examine the complexities of these linkages. Our findings suggest that existing safety thresholds, predominantly based on single-element studies, would need to be modified to account for the joint exposure effect from heavy metals and other environmental pollutants. Secondly, considering that the health effects may worsen as the child grows due to the cumulative impact of pre-and postnatal exposures, there is likely a need to establish community–based early childhood development interventions in areas with ASGM activi-ties. The need to provide young children with cognitively stimulating activities and expose them to learning opportuni-ties and better activity spaces is crucial. The current study provides evidence that supports a call for action to reduce

**Table 4. Marginal effects of individual chemicals and associated covariates on neurodevelopmental abilities.**

| Prenatal Exposure Characteristics | General Impairment | Social Interactions | Language Ability | Gross Motor | Fine Motor |
|---|---|---|---|---|---|
| | aPR | aPR | aPR | aPR | aPR |
| *(Intercept)* | *0.584* | *0.008* | *0.145* | *1.584* | *0.507* |
| Pb | 0.762* | 1.1998 | 0.509*** | 0.775* | 0.868 |
| As | 0.984 | 0.999 | 0.961 | 0.930 | 0.964 |
| Cd | 1.125 | 1.322 | 0.866 | 1.095 | 1.182 |
| Hg | 0.844 | 0.683 | 0.781† | 0.833† | 0.878 |
| *Sex* | | | | | |
| Boys | Ref | | | | |
| Girls | 0.607* | 0.716 | 0.927 | 0.737† | 0.868 |
| *Birth Weight* | *0.819* | *1.122* | *0.903* | *0.627* | *0.993* |
| *Maternal Education* | | | | | |
| No education | Ref | | | | |
| Primary school | 0.807 | 1.196 | 0.837 | 0.730 | 0.605 |
| Ordinary level | 0.308** | 0.426 | 0.158† | 0.202** | 0.461 |
| Tertiary | 0.736 | 4.438E-07 | 1.68E-06 | 0.863 | 0.829 |
| *Child age* | *1.061*** | *1.067* | *1.052* | *1.054** | *1.030* |
| *SES status* | | | | | |
| High | Ref | | | | |
| Moderate | 0.799 | 0.985 | 1.711 | 0.878 | 0.735 |
| Low | 0.495† | 0.418 | 1.345 | 0.565 | 0.373* |

**Note:**

†=significant at 0.10;

*=significant at 0.05;

**=significant at 0.001;

***=significant at 0.0001

**Table 5. Individual effects from the covariate-adjusted univariate model on neurodevelopmental abilities.**

| General Impairment | | Social | Language | Gross Motor | Fine Motor |
|---|---|---|---|---|---|
| | aPR | aPR | aPR | aPR | aPR |
| Pb | 0.895** | 1.183 | 0.561*** | 0.885* | 0.973 |
| As | 0.981* | 1.0096 | 0.916 | 0.952 | 0.964 |
| Cd | 1.014 | 1.219 | 0.777* | 1.000 | 1.066 |
| Hg | 0.930† | 0.784 | 0.459† | 0.912† | 0.935 |

**Note:**

†=significant at 0.10;

*=significant at 0.05;

**=significant at 0.001;

***=significant at 0.0001

environmental pollution associated with ASGM activities in affected areas. There is a need to promote the implementation of early childhood stimulation programs to minimize the impact of developmental impairments among children residing in ASGM areas.

The findings that the individual marginal effects of Pb and Hg on neurodevelopmental abilities are amplified beyond what would be expected under the additive assumption when reacted with other chemical elements calls for strengthening the growing demand to modify existing health risk reduction strategies, which are predominantly based on single-element assessment. Adapting existing safety thresholds in the context of multi-chemical exposures, which is a reality for women and children in gold mining areas, may be timely. While the combined effects of these heavy metals may lead to synergistic impacts, socioeconomic disparities, and gender likely play an interactive role. This underscores the need to incorporate subgroup modeling into the multi-chemical exposure assessment framework.

## 5. Limitations of the study

This study is limited to prenatal exposure among children born to mothers in ASGM communities. This follow-up nested study does not consider exposures to chemical elements post-delivery. The actual amount of the toxic chemical elements transferred to the fetus may vary from one woman to another depending on maternal health status [62]. Future bio-monitoring studies should consider maternal health status and individual children's exposure status. Recruitment was limited for consistency to mother-child pairs who stayed in the ASGM area to follow up on neurodevelopment. Children were assessed at a time when they were not sick or in need of any medical assistance. The individual children's percentile scores in each domain during the assessment, apart from the passing or failing of a child at the 90% percentile level for their age, might have reduced the precision of the results. However, using a validated African-developed tool (i.e., the MDAT) provides robustness to the findings and accounts for the local settings. The adequate sample size and sampling of appropriate bio-indicators (i.e., maternal prenatal blood for Hg, Pb, and Cd; and urine for As) [62] provide value to the integrity of the findings.

Although this study controlled for factors including birthweight, socioeconomic status, gender, education, and age of the child, a multitude of other environmental exposures and socioeconomic factors can contribute to or mediate cognitive impairment. As a result of the complexity of contributions to cognitive impairment, even though this current research documents an association, the results are not causal. Instead, the results strongly suggest a need to focus on multiple chemical interactions when seeking to understand the influence of environmental exposures on cognitive impairment. Further, future studies would benefit from improved occupational classifications and direct exposure biomarkers.

## 6. Conclusion

Children born to women in areas with ASGM activities who are prenatally exposed to Pb, Hg, and As are at increased risk for neurodevelopmental impairment. The cumulative impact of all chemicals was significant for gross motor, language ability, and general impairment. The magnitude of neurodevelopmental impairments increases as children grow. The independent effects of Pb and Hg were amplified beyond what would be expected under the additive assumption, suggesting synergistic or potentiating effects. Neurodevelopmental health risks among children varied according to gender, educational, and socioeconomic status (SES), drawing attention to the need for studies focused on environmental health inequities in the African context.

## Supporting information

**S1 Table** Forward-backward stepwise regression performance metrics and variables maintained. The blue-colored variables are finally selected according to AIC and Deviance.
(DOCX)

**S2 Table. Median heavy metal levels by mother's occupation.**
(DOCX)

**S1 Data. De-identified data and detailed information regarding the participants.**
(SAV)

**S1 Checklist. STROBE Statement—checklist of observational studies.**
(DOCX)

**S2 Checklist. Inclusivity in global research.**
(DOCX)

## Acknowledgments

The authors would like to thank all the healthcare workers, women, and children participating in the *Tanzania Mining and Health Cohort* for participating in this study. The authors acknowledge support and assistance from the Strengthening One Health and Planetary Health in East Africa (SOPHEA).

## Author contributions

**Conceptualization:** Elias C. Nyanza, Raphael J. Mhana, Deborah S.K. Thomas, Agapiti P. Kisoka.

**Data curation:** Elias C. Nyanza, Raphael J. Mhana, Agapiti P. Kisoka.

**Formal analysis:** Elias C. Nyanza, Moses Asori, Deborah S.K. Thomas.

**Funding acquisition:** Elias C. Nyanza.

**Investigation:** Deborah S.K. Thomas.

**Methodology:** Moses Asori.

**Software:** Moses Asori.

**Supervision:** Elias C. Nyanza, Deborah S.K. Thomas, Agapiti P. Kisoka.

**Writing – original draft:** Elias C. Nyanza.

**Writing – review & editing:** Elias C. Nyanza, Raphael J. Mhana, Moses Asori, Deborah S.K. Thomas.

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
