## [Decision Letter · Decision Letter 0]

9 Feb 2025

PGPH-D-24-02727

Effects of Prenatal Lead, Mercury, Cadmium, and Arsenic Exposure on Children’s Neurodevelopment in an Artisanal Small-scale Gold Mining Area in Northwestern Tanzania Using a Multi-chemical Exposure Model

Dear Dr. Elias C. Nyanza,

Thank you for submitting your manuscript to PLOS Global Public Health. After careful consideration, we feel that it has merit but does not fully meet PLOS Global Public Health’s publication criteria as it currently stands. Therefore, we invite you to submit a revised version of the manuscript that addresses the points raised during the review process.

We look forward to receiving your revised manuscript.

Kind regards,

Muhammad Asaduzzaman, MD MPH MPhil

Academic Editor

Journal Requirements:

2. We do not publish any copyright or trademark symbols that usually accompany proprietary names, eg (R), (C), or TM  (e.g. next to drug or reagent names). Please remove all instances of trademark/copyright symbols throughout the text, including ® on page 5.

Reviewers' comments:

Reviewer's Responses to Questions

**Comments to the Author**

1. Does this manuscript meet PLOS Global Public Health’s publication criteria ? Is the manuscript technically sound, and do the data support the conclusions? The manuscript must describe methodologically and ethically rigorous research with conclusions that are appropriately drawn based on the data presented.

Reviewer #2: Yes

Reviewer #3: Yes

Reviewer #4: Yes

2. Has the statistical analysis been performed appropriately and rigorously?

Reviewer #2: Yes

Reviewer #3: Yes

Reviewer #4: Yes

3. Have the authors made all data underlying the findings in their manuscript fully available (please refer to the Data Availability Statement at the start of the manuscript PDF file)?

Reviewer #2: Yes

Reviewer #3: Yes

Reviewer #4: No

4. Is the manuscript presented in an intelligible fashion and written in standard English?

Reviewer #2: Yes

Reviewer #3: Yes

Reviewer #4: Yes

5. Review Comments to the Author

Reviewer #2: Overall this is a great paper and I have only minor comments. I also applaud this research as it pertains specifically to more marginalized populations and since I spent 15 years setting up medical services in remote rural underserved settings, I think of folks like these as exactly the kind of populations I like to see supported. I only have some minor comments:

How did you control for cognitive impairment related to other potential causes, since diet and many, many other causes are also known to contribute to this? The inconsistency you mention in previous studies probably is impacted by this issue.

You mention "Children developed additional impairments at three to four years, providing evidence that neurodevelopmental deficits from prenatal exposures continue 1 later in life" but although I would believe that, this seems like an assumption that does not consider the many other factors that can contribute to deficits besides pre-natal exposure, such as early childhood education, mental stimulation at home, diet in infancy and early childhood, etc. Not so much proof, as you indicate, but an association that certainly should be explored further to try and demonstrate a causal relationship in a study that controls for other factors besides heavy metal exposure. I have no doubt such a study would in fact find that independent of all other factors, the heavy metal exposure carries a greatly increased risk, but I think that's something to explore in future studies. You do actually mention the need for this in the very last sentence of the discussion, but I think you may need to go through the paper and be careful to acknowledge the likely multifactorial causes that are surely at work here in addition to chemical exposure in utero--especially since this entire paper is about the synergistic multifactorial impact of heavy metal exposure, where previous studies have mainly looked at single or few possible contaminant exposures. Surely the other external factors in utero, infancy and early childhood also have potentially synergistic impacts on this as well? Maternal diet? Birth complications? Maternal and infant/early childhood infections? Breastfeeding and early infant/childhood diet? Exposure to other industrial or agricultural products that may be known (or suspected) to have an impact?

The gender differences seem to indicate that either there is some gender-based difference in development in utero that creates additional risk, OR perhaps there are gender-based differences in exposure in infnacy and early childhood; I think the study needs to acknowledge this as a limitation.

Reviewer #3: In the introduction, the authors state that: 'An estimated 200 million children under five in Sub-Saharan Africa do not reach their full cognitive development potential.' They need to specify whether this figure refers to an annual estimate, a decade, or another time frame.

Reviewer #4: The manuscript is well-written in standard English with very little typos (only one seen). The structure of the manuscript meets PLOS requirement with the contents under each heading adhering to acceptable international standards. The tables and figures are labeled and explained to my satisfaction. Though I did not have access to the data, adequate analysis was performed to answer the aim of the study and there is a request to perform one or two additional analysis.

6. PLOS authors have the option to publish the peer review history of their article (what does this mean? ). If published, this will include your full peer review and any attached files.

**Do you want your identity to be public for this peer review?** For information about this choice, including consent withdrawal, please see our Privacy Policy .

Reviewer #2: **Yes: ** Dr. Benjamin LaBrot

Reviewer #3: **Yes: ** Rita Nugem

Reviewer #4: No

---

## [Editor Report · Decision Letter 1]

10 Apr 2025

Effects of Prenatal Lead, Mercury, Cadmium, and Arsenic Exposure on Children’s Neurodevelopment in an Artisanal Small-scale Gold Mining Area in Northwestern Tanzania Using a Multi-chemical Exposure Model

PGPH-D-24-02727R1

Dear Dr. Elias C. Nyanza,

We are pleased to inform you that your manuscript 'Effects of Prenatal Lead, Mercury, Cadmium, and Arsenic Exposure on Children’s Neurodevelopment in an Artisanal Small-scale Gold Mining Area in Northwestern Tanzania Using a Multi-chemical Exposure Model' has been provisionally accepted for publication in PLOS Global Public Health.

Best regards,

Muhammad Asaduzzaman, MD MPH MPhil

Academic Editor